

# Co-observation of strongly convective precipitation using VHF atmospheric radar and dual-polarized microwave radiometer during a typhoon passage

Shih-Chiao Tsai[1], Yen-Hsyang Chu[2], Jenn-Shyong Chen[3]

[1]Department of Environmental Information and Engineering, National Defense University, Taoyuan, Taiwan
[2]Graduate Institute of Space Science and Engineering, National Central University, Taoyuan, Taiwan
[3]Center for General Education, China Medical University, Taichung, Taiwan

*Correspondence to*: Jenn-Shyong Chen (james.chen@mail.cmu.edu.tw)

**Abstract.** The Chung-Li very-high-frequency (VHF at 52 MHz) atmospheric radar and a dual-polarized microwave radiometer were operated collaboratively to investigate strongly convective precipitation while the typhoon Trami just passed through the Taiwan in Aug, 2013. First, respective Doppler velocities of clear-air and precipitation echoes were identified automatically by the VHF radar as clearly as possible. Two approaches were designed for this purpose: contour-based and peak-finding processes. The two approaches initially determined some major spectral centers or peaks, which were usually redundant, and then proper sifting and clustering were performed for the redundant spectral centers or peaks to yield several mean locations of Doppler velocities for profiling. The outcomes of the two approaches were consistent in general. With the estimated Doppler velocities, a tracing process was developed for Doppler profiling, in which Doppler velocity shear was one of the significant criteria in the tracing process. The radar echoes collected by the VHF radar during rainy and strongly convective atmosphere have been investigated to validate the two approaches and the tracing process. About 98% of the tracings could identify the Doppler profiles of clear air and precipitation, even the atmosphere was disturbed severely. The radar spectral parameters, Doppler profiles, and the information from a dual-polarized microwave radiometer as well as the simulation of weather model, were examined jointly. It signified that strong updraft and turbulent atmosphere could bring the liquid water to the height above the melting layer, and then the Bergeron effect and coalescence process on formation of ice crystal and graupel above the height of the melting layer occurred accordingly.

## 1 Introduction

Atmospheric radars operated at very-high-frequency (VHF) and ultra-high-frequency (UHF) bands are powerful instruments in remote sensing of the atmosphere from troposphere to ionosphere. In addition to the clear-air turbulence and plasma irregularities, precipitation is also a significant source of the radar echoes, which has been extensively investigated for decades (e.g., Lucas et al. 2004; McDonald et al. 2006; Kanofsky et al. 2008; Chu et al. 2008; Su et al. 2009; Chen et al., 2020).

Wind velocity is a basic measurement of using VHF/UHF atmospheric radars. Owing to radio interference echoes or unwanted signals such as those from airplanes, birds, radio frequencies, and so on, the wind velocity may be estimated





erroneously. Moreover, the ground clutter that produces significant peak in the Doppler spectrum around zero velocity could be a problem for some radars. A simple way to mitigate the influence of ground clutter is via direct-current (DC) removal in the raw data and/or average of both sides of the spectral lines around zero velocity. For more effectively in removing the contamination of ground clutter, however, several sophisticated algorithms have been developed, e.g., neural network

(Clothiaux et al. 1994), Genetic algorithm (Chen et al. 2002), Fuzzy logic-based methods (Cornman et al. 1998; Morse et al. 2005). After supressing the signals of ground cutters or other interferences, tracing the Doppler profiles of the echoes throughout the height is more applicable. For example, Anandan et al. (2005) proposed an adaptive moments estimation technique and Sinha et al. (2017) used a muiltiparamter cost function method to trace the Doppler profile of clear air motion in low signal-to noise ratio (SNR) condition, in which the wind-shear value was an important criterion in tracing.

The aforementioned algorithms and tracings focused on extracting the mean Doppler velocity of clear-air motion. In case of strong hydrometeor echoes that are closer to the clear-air echoes in the Doppler spectrum, Gan et al. (2015) proposed a two-echo method to exclude the hydrometeor or precipitation echoes so that the spectral parameters of clear air could be estimated in more accurate. The precipitation echoes, on the other hand, are useful for estimating the drop size distribution (DSD) of falling hydrometers, and DSD is an essential measurement of deducing the rainfall rate, liquid water content, terminal velocity

of hydrometer, and so on (Atlas et al. 1973; Brandes et al. 2004). Therefore, the clear air echoes should be removed from the observed Doppler spectrum for extraction of precipitation echoes, and several methods have been explored for this purpose. For example, deconvolution operation (Lucas et al. 2004; Rajopadhyaya et al. 1993), spectral fitting (Wakasugi et al. 1986; Sato et al. 1990; Kobayashi et al. 2005), cluster analysis (Williams et al. 2000), peak-searching process for the air spectrum (Campos et al. 2007), and others based on these methods (Rajopadhyaya et al. 1999; Schafer et al. 2002).

To the authors' knowledge, there seems lack of an exclusive and effective process available to concurrently identify and trace clear-air and precipitation echoes in the spectral domain. In view of this, we intend to propose a new tracing process of Doppler profiling of clear air and precipitation echoes, and obtain the required Doppler parameters for further study. To this end, two approaches of identifying the spectral peaks of clear air and precipitation were developed, that is, contour-based and peak-finding approaches. The proposed approaches and tracing process were demonstrated for a convective atmosphere

detected by the Chung-Li VHF radar. A convective atmosphere yields a circumstance of wide variation in Doppler velocities of clear air and precipitation, which is an excellent test-bed of our approaches. Moreover, a collaborative observation of the VHF radar and a dual-polarization microwave radiometer for convective precipitations were carried out to show a potential application of these approaches and tracing process. The information of raindrop shape and liquid water content measured by the radiometer, combined with the VHF radar measurement, were used to investigate the interrelation between hydrometeor

particles and clear-air turbulence in a strongly convective atmosphere.

This paper is organized as follows. In Sect. 2, the two approaches of estimating the respective Doppler/spectral parameters of clear air and precipitation are described. Section 3 presents the spectral parameters: mean Doppler velocity and spectral width. Comparisons between the spectral parameters obtained from the two approaches are also made in this section. Section 4 addresses the process to trace the Doppler profiles in convective atmospheric condition including heavy rain. Advantages and



limitations of the tracing process will be discussed. Finally, co-observations of precipitation with VHF radar and dual-polarized microwave radiometer are shown in Sect. 5, where the spectral parameters, estimated from the proposed approaches, and the hydrometeor parameters, observed by the radiometer, are jointly discussed. Conclusions are drawn in Sect. 6.

## 2 Estimates of Doppler parameters

A flowchart that briefly describes the procedures of estimating spectral parameters with the contour-based and peak-finding approaches is shown in Fig. 1. Before practicing these approaches, the radar data were processed in the following ways. First, Doppler spectra of the radar echoes were produced by using fast Fourier transform (FFT) with 32, 64 and 128 raw data pints, respectively, in which the direct-current (DC) component has been removed. We then performed incoherent integrations with the times of 4, 2 and 1, respectively, for the 32-, 46- and 128-point Doppler spectra in frequency domain to render the Doppler spectra with the same duration (~17 s). The 3-, 7-, and 13-point equal-weight running mean were, respectively, applied to 32-, 64- and 128-point integrated Doppler spectra for smoothing. After this procedure the contour-based and peak-finding approaches were implemented to the smoothed spectra to estimate the Doppler velocities of various targets and their respective spectral widths. The details of the two approaches are described below.

### 2.1 Contour-based approach

For clarity, the approach is stated step by step:

1)  *Create 2-D spectral map*

The original 1-D Doppler spectrum was first extended intentionally to another orthogonal spectral dimension with a Gaussian function for each spectral line to construct a modelled 2-D Doppler spectrum, in which the height of the spectral line and one eights of the FFT number (e.g., 4 for 32-point FFT) were, respectively, assigned to the peak value and the standard deviation of the Gaussian function.

2)  *Obtain contour lines*

The built-in function C=contour(S, n) in the MATLAB software was employed to obtain the contour lines of the modelled 2-D spectrum, where S is the matrix of the 2-D spectrum in linear scale, n is the contour levels, and C is the matrix containing the coordinates of the resultant contour lines (Fig. 2). The contour levels were given to be adaptive to the maximum value ($M_0$) of the spectral lines in accordance with a relation given below:

$$N_{level} = 2 + floor[10\log_{10}(M_o)] \tag{1}$$

where "*floor*" is the MATLAB function that rounds the value to the nearest integer less than or equal to that value. As a result, the number of contour levels is larger for a higher $M_0$ so that some minor contour centers can also be identified for later investigation.





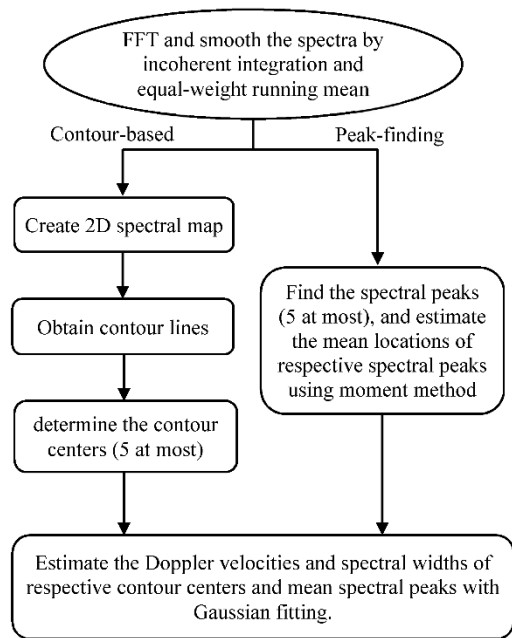


**Figure 1: Flow chart of contour-based and peak-finding approaches to determination of Doppler velocity and spectral width.**

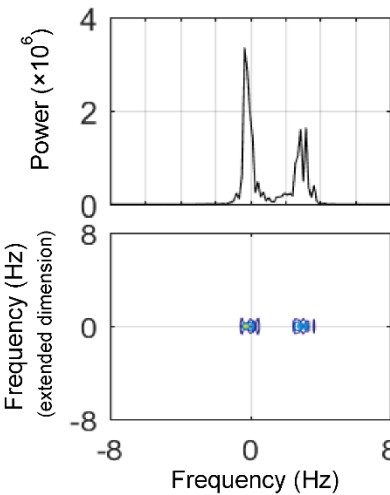

**Figure 2: (top) Original Doppler spectra, (bottom) extended two-dimensional Doppler spectra presented in contour lines.**


*3)   Determine the contour centers*

    With the coordinates of contour lines, C, the contour centers were determined by applying the locating process proposed by Chen et al. (2008). The determined contour centers will be nearly aligned in the dimension of the original 1-D Doppler spectrum because the modelled Doppler spectrum in the orthogonal dimension processed in step 1 hardly biases the locations





of the contour centers. Therefore, in the following statement the location of contour center denotes the coordinate of the contour center in the original Doppler spectral dimension.

Five contour centers that are most significant were selected at most through the process mentioned above, but the ones at the lowest contour level were discarded. Then the identified contour centers within a velocity interval ($v_{interval}$) were grouped to calculate a weighted mean center, where $v_{interval}$ was assigned to 1.4 m/s and the respective contour levels of the contour

centers were used as the weights in calculation of the weighted mean center (mean center hereafter).

At most 5 mean centers can be obtained in each range gate if the separation of the adjacent contour centers are larger than $v_{interval}$ and the ratio of the spectral power, averaged from the spectral lines centered on the mean center, to a referenced noise level is higher than a specified threshold value ($R_p$), say, 0.75. Giving a higher $R_p$ will reject more minor mean centers. In this study, the referenced noise level was estimated from the mean of the lowest one fourths spectral power lines, and the averaged

spectral power of each mean center was calculated from 6, 10, and 14 spectral lines centered on the mean center, respectively, for the Doppler spectra of 32, 64, and 128-point FFTs.

*4)   Estimate the Doppler velocities and spectral widths of respective targets with Gaussian fitting*

Once the mean centers were obtained, for each mean center we fitted a Gaussian function (Gaussian fitting thereafter) to the original 1-D Doppler spectrum with 5, 7, 9, and 13 spectral lines centered on the mean center, respectively. Among the four

Gaussian fittings, the Gaussian function with minimum standard deviation and the separation between the Gaussian-fitted peak and the mean center less than a half of the velocity interval ($v_{interval}$=1.4 m/s) was adopted. Finally, the Gaussian-fitted peak denotes the Doppler velocity of the mean center, and the standard deviation is the corresponding spectral width. At most 5 pairs of Doppler velocities and spectral widths can be retained for later investigation if 5 mean centers are identified in the third step. In addition, the normalized amplitudes of the Gaussian fitting were retained for future statistical analysis and

Doppler profiling.

**2.2 Peak-finding approach**

Peak-finding is commonly employed in several of the aforementioned algorithms for estimate of Doppler velocity and profiling. We have developed our own process as follows:

*1)   Find the spectral peaks*

This step is similar to the third step of the contour-based approach, except that the function "findpeaks" built in MATLAB was used to find 5 most significant spectral peaks that were separated by at least a velocity interval ($v_{interval}$). Note that $v_{interval}$ was given as that in the contour-based approach. In addition, the spectral peaks at the lowest contour level, as given in the contour-based approach, were discarded.

For each spectral peak, a mean location was further calculated by the moment method with several spectral lines centered on the peak. The numbers of spectral lines used in the calculation of mean location were 6, 10, and 14 for the Doppler spectra

estimated from 32, 64, and 128-point FFT, respectively. Such estimate of mean center for each spectral peak is to provide a





more representative mean location for the Gaussian fitting executed in the following step. Finally, the calculated mean location was discarded when the ratio of the averaged spectral power to the referenced noise level was smaller than the threshold value, $R_p$. Here the averaged spectral power, referenced noise level, and $R_p$ were calculated as those in the third step of the contour-based approach.


*2) Estimate the Doppler velocities and spectral widths of respective targets with Gaussian fitting*

The Gaussian fitting performed in the fourth step of the contour-based approach was executed for the respective mean locations of spectral peaks to provide several pairs of mean Doppler velocity and spectral width. After this fitting, we calculated another pair of mean and standard deviation additionally via moment method with the spectral lines centered on the Gaussian-fitted mean Doppler velocity; the purpose of this extra calculation is to make a further comparison with the contour-based and peak-finding approaches. The spectral lines included in the extra calculation were within 2 times of the Gaussian-fitted standard deviation on both sides of the Gaussian-fitted mean Doppler velocity. As a result, at most 5 pairs of mean and standard deviation can be obtained additionally if the Gaussian fitting produces 5 results initially.

It should be clarified here that redundant Doppler velocities and spectral widths, obtained from the two locating approaches and the additional moment-method calculation, are to acquire the spectral parameters as complete as possible; that is beneficial for Doppler profiling later.

## 3 Validation of spectral parameters

To validate the approaches described in Sect. 2, the Chung-Li VHF radar data collected on 21 Aug, 2013, were examined. Heavy convective precipitation was prevailing during the observational period while the typhoon Trami just passed through the Taiwan area. The radar parameters were set as follows: vertically transmitted radar beam with multiple carrier-frequency mode (i.e., the carrier frequencies of 51.5, 51.75, 52, 52.25, and 52.5 MHz were transmitted alternately), inter-pulse period 200 μs, number of coherent integrations 128, pulse width 1 μs, sampling interval 1 μs (corresponding to a range resolution of 150 m), lowest sampling height 1.5 km, number of range gates 80, and without pulse coding. As a result, the sampling time of radar returns for each carrier frequency was 0.128 s (=200 μs×128 integration times×5 frequencies). For the goal of this study, however, only the echoes from the carrier frequency of 52 MHz were analyzed.

Fig. 3a shows the height-time distribution of echo intensity and Fig. 3b presents time series of the disdrometer-observed rainfall rate. As indicated, the primary radar returns in region 1 were characterized by a stratified and shallow distribution with height, which were associated with weak precipitation. By contrast, those in region 2 accompanied with a very intense rainfall and extended up to 5 km more. Note that the radar data shown in Fig. 3a have been examined by Chen et al. (2016) for evaluation of range imaging technique (a multiple-frequency analysis). These radar data were used again in our study for demonstrating the effectiveness of the Doppler profiling process. Furthermore, an investigation of raindrop shape and the effect





of convective atmosphere on the composition of hydrometeor particles was also made with the information of microwave radiometer and the present radar data.

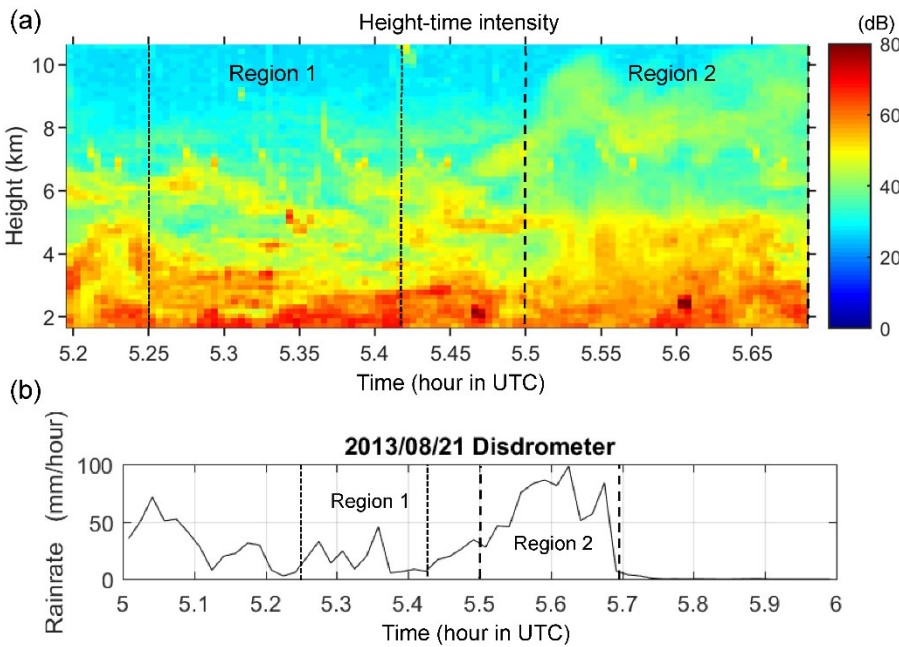

**Figure 3: (a) Height-time intensity plot of VHF atmospheric radar echoes. Little precipitation occurred in the region 1, but in the region 2 precipitation echoes are plentiful and strong below 6 km, and intense echoes were also observed above 7 km. (b) Rain rate detected by a disdrometer located near the VHF radar site.**

Fig. 4a and b shows two examples of the Doppler spectra (colored image) calculated from 64-point FFT for the radar echoes in the regions 1 and 2, respectively, where positive (negative) Doppler velocity means downward (upward). The number of incoherent integration of Doppler spectrum was 2, resulting in a duration of about 17 s for a Doppler spectrum. Notice that the positive Doppler velocity was extended to about 16 m/s, which exceeded the Nyquist Doppler velocity of about 11 m/s. This was done by copying the one fourths spectra on the leftmost side to the rightmost side, making the precipitation spectra more complete when the falling velocity of the precipitation was larger than the Nyquist Doppler velocity in observation. This is beneficial for locating the spectral peaks of precipitation at larger Doppler velocities. The copying, nevertheless, is not always necessary if the sampling time is short enough so that the Nyquist Doppler velocity is much larger than the Doppler velocity of precipitation.

Also shown in Fig. 4a and b are the mean Doppler velocities and the spectral widths of clear-air and precipitation echoes, which were estimated from the contour-based (marked in red) and peak-finding (marked in black) approaches, respectively. In Fig. 4a, only the clear-air echoes were visible, while the clear-air (distributed around 0 m/s) and precipitation (characterized by salient positive Doppler velocities) echoes were both present in Fig. 4b. As seen, the spectral parameters estimated by the

two approaches were generally in agreement, although there were still some minor discrepancies in them. For a clearer
inspection, Fig. 4c displays the spectral power in curve for several sampling gates for comparison. As seen in the upper panel
of Fig. 4c, where shows the motion of clear air, the peak-finding approach determined four Doppler velocities at the first gate
(black dots), but the contour-based approach provided only one (red dot). Apparent discrepancies in the calculated Doppler
velocities can also be found at the heights around 8.4 km, 8.7 km, and 9.2 km in the upper panel, and at around 4.8 km in the
lower panel of Fig. 4c. In view of this, a further classification and grouping of these Doppler velocities are necessary for tracing
the Doppler profiles automatically; this will be addressed in the next section.




**Figure 4: Comparison between the locating results of contour-based and peak-finding approaches. (a) Clear-air radar echoes,
exclusive of the airplane echoes above the height of 9 km; (b) coexisted radar echoes of clear air and precipitation. The spectral
power is displayed in dB. Red dot represents the Gaussian-fitted Doppler velocities determined by the contour-based approach.**
**Black dot and open circle denote, respectively, the Gaussian-fitted and moment-estimated Doppler velocities in the peak-finding
approach. The length of the bar denotes full spectral width. (c) Self-normalized spectral power represented in linear scale (curve),
where dot, circle, and bar length have the same definitions as those in (a) and (b).**

Considering that different FFT data lengths yield different Doppler spectral resolutions and may result in different locations
and numbers of spectral peaks, comparisons of the distributions of Doppler velocities and spectral widths calculated by 32-,
64-, and 128-point FFT were made and shown in Fig. 5, where the calculations from the contour-based approach are shown
($V_C$: Doppler velocity, $SW_C$: spectral width). For the clear air echoes shown in Fig. 5a, the three distributions of $V_C$ and $SW_C$
were relatively narrow, nearly invariant with height, and very close to each other, except for the height above ~8 km where





the signal to noise ratio (SNR; indicated by thick black curve) was much lower than 10 dB (denoted by the dotted vertical line). Here the SNR was calculated from the 64-point data length and self-normalized for presentation. By contrast, the distribution of $V_C$ shown in Fig. 5b highly varied with height and $SW_C$ was relatively broad.

A noticeable feature in the distributions of $SW_C$ in Fig. 5b was a bias of distribution peaks to smaller values as the FFT point increased. A detailed examination showed that in the environment of intense precipitation, there was a tendency for the mean of spectral width to become lager (smaller) if a FFT with less (more) data points was performed to generate the Doppler spectra, as shown in the left panel of Fig. 6b. As seen, the mean of $SW_C$ was the largest for 32-point FFT (black thin curve), and the smallest for 128-point FFT (blue curve) throughout the range. Such feature can be attributed to the smoothing effects on the spectral lines caused by the windowing function in time domain used to perform FFT and the practice of the incoherent integration in spectral domain. By contrast, the mean of $SW_C$ shown in Fig. 6a, where the clear-air echoes dominated, were nearly the same. This is expected because in region 1 the atmosphere with little precipitation aloft was relatively stable, resulting in narrow spectral width. On the other hand, irrespective of region 1 or region 2, the standard deviations of $SW_C$ were essentially independent of the point of FFT used in the calculation of Doppler spectrum, as shown in the right panels of Fig. 6a and b, suggesting the robustness of the process in the determination of the spectral width.

A further comparison between the mean Doppler velocities and spectral widths obtained from the contour-based and peak-finding approaches are shown in Fig. 7, in which $V_C$ and $SW_C$ are the Doppler velocity and spectral width of the contour-based approach, $V_{pg}$ and $SW_{pg}$ are the corresponding results of Gaussian fitting in the peak-finding approach, and $V_{pm}$ and $SW_{pm}$ denote the additional estimates using the moment method in the peak-finding approach. The histogram in each panel shows the distribution of the difference between the two spectral parameters compared, which is shown between -2 and 2 m/s. As shown in Fig. 7a for clear air echoes, no matter what the methods were used, the estimated spectral parameters were in good agreement, although the slope (~0.96) of the regression line for the values of $SW_C$ and $SW_{pm}$ (at the second row and middle column) was slightly lower than the others (close to 1). Note that, for the moment method executed in the peak-finding approach, the spectral lines within ±2 times of the standard deviation, obtained from the Gaussian-fitting process and centered on the mean Doppler velocity, were used in the calculation, which included more spectral lines than those of the Gaussian fitting method. As a result, the standard deviations (i.e., spectral widths) estimated from the moment method could be larger than those from the Gaussian-fitting method. Such situation could be worse under the conditions of strongly turbulent atmosphere and precipitation. As shown in Fig. 7b, the slopes of the regression lines for the values of $SW_C$ and $SW_{pg}$, $SW_C$ and $SW_{pm}$, and $SW_{pg}$ and $SW_{pm}$ were smaller than those in Fig. 7a, which reduced to about 0.98, 0.81, and 0.95, respectively. Moreover, the scatter dots in Fig. 7b spread out more than in Fig. 7a.



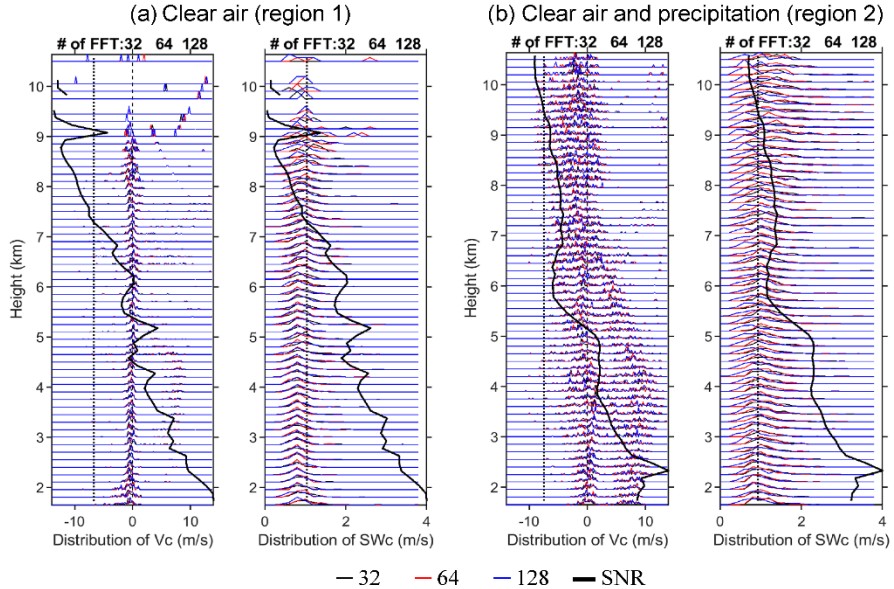

**Figure 5: Comparison between the spectral parameters obtained by using 32 (thin black curve), 64 (red curve) and 128 (blue curve) FFT points, respectively. V$_C$ and SW$_C$ denote the Doppler velocity and spectral width determined by the contour-based approach. (a) Clear-air echoes with little precipitation (the region 1 in Fig. 3a); (b) coexisted clear-air and precipitation echoes (the region 2 in Fig. 3a). Thick profiling curves are the normalized SNR obtained from the 64-FFT point calculation, normalized between 0 and maximum value in dB. Dotted vertical line denotes the SNR level at 10 dB.**

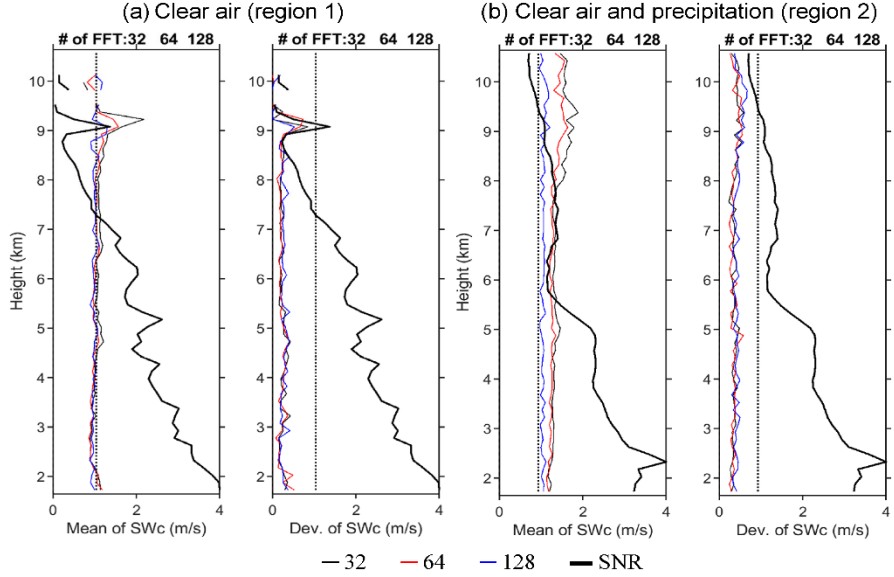

**Figure 6: Statistical comparisons between the spectral widths obtained by using 32 (thin black curve), 64 (red curve) and 128 (blue curve) FFT points. (a) and (b) uses the spectral widths given in Fig. 5a and b, respectively.**

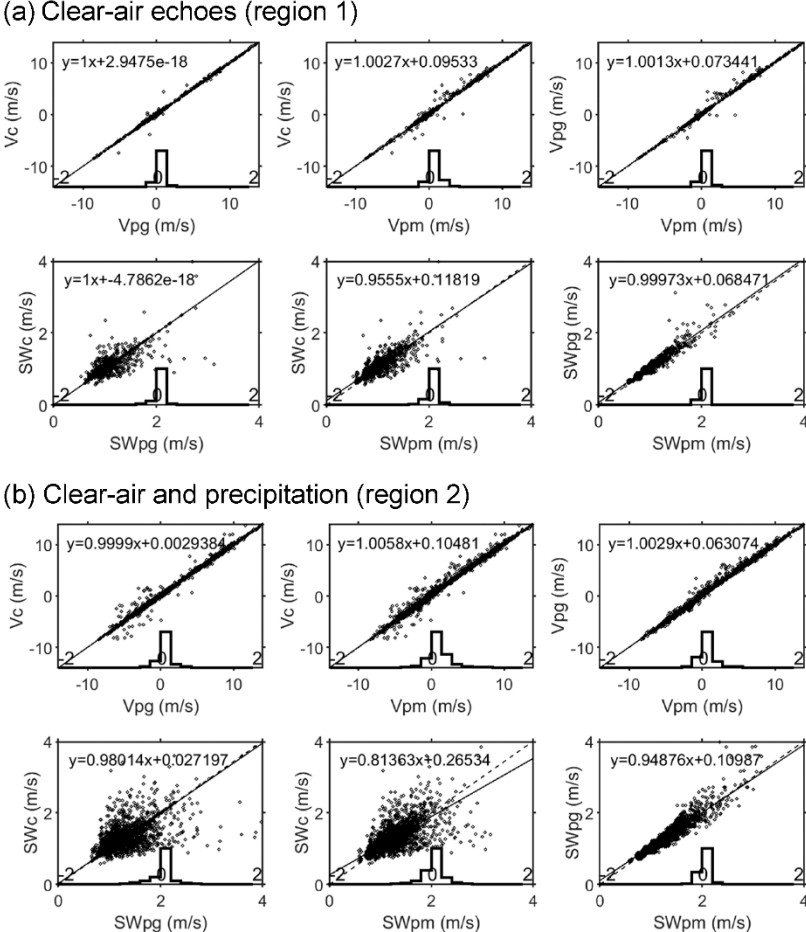

**Figure 7: Comparison between the spectral parameters obtained from the contour-based and peak-finding approaches. (a) and (b)**
**show the clear-air echoes in region 1 and the coexisted clear-air and precipitation echoes in region 2, respectively. V$_C$ (S$_{WC}$), Vpg**
**(SWpg), and Vpm (SWpm) represent the Doppler velocity (spectral width) resulting from contour-based approach, and Gaussian-**
**fitting and moment method in the peak-finding approach, respectively. The histogram represented by step curve (self-normalized)**
**in each panel shows the difference between the two parameters compared, ranging from -2 and 2 m/s.**

To estimate respective spectral widths of clear air and precipitation, we categorized the spectral widths into two different
groups in terms of mean Doppler velocity: between -3 and 3 m/s, and larger than 3 m/s. This is because that most of the Doppler
velocities larger than 3 m/s were found to result from precipitation echoes. Fig. 8 shows the results calculated from the spectral
widths (SW$_C$) of the contour-based approach for the radar echoes in the region 2 of Fig. 3a. In Fig. 8a, all the outcomes of SW$_C$
were used to estimate the mean ($\mu_{SWc}$) and standard deviation ($\sigma_{SWc}$) of SW$_C$. It is apparent that the mean of spectral widths of
precipitation echoes (red solid curve) were slightly larger than those of clear-air echoes (blue solid curve) below ~6 km; notice
that strong precipitation occurred mainly below ~6 km in the region 2 of Fig. 3a. Even if the outcomes having the Gaussian-

fitted amplitudes larger than 0.5 or 0.8 (a normalized value) were adopted only, the mean of spectral widths of precipitation echoes were still larger than those of clear air echoes below ~6 km, as demonstrated in Fig. 8b-c. Moreover, the deviations ($\sigma_{SWc}$) of spectral widths were small for clear-air and precipitation echoes both. In view of this, the precipitation echoes indeed

265    distributed over a broader velocity interval than the clear-air echoes. This result is expected because the precipitation was accelerated from rest to a terminal velocity of about 10 m/s in the vertical direction, and the falling velocities of precipitation spread in a wider velocity interval due to different raindrop sizes.

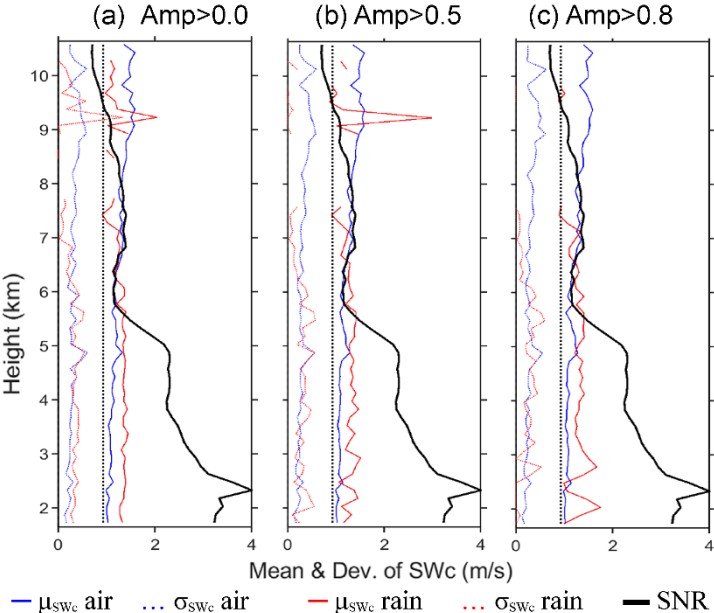

270    **Figure 8: Comparisons between the spectral widths in two Doppler velocity intervals: between -3 and 3 m/s (blue curve), and larger than 3 m/s (red curve). The outcomes in the region 2, obtained from the contour-based approach using 64 data points in FFT, are shown. Thin and light-dotted curves represent, respectively, the mean ($\mu_{SWc}$) and standard deviation ($\sigma_{SWc}$) of spectral width. Thick solid curve displays the mean SNR normalized between 0 and the maximum value in dB, and dotted vertical line denotes the SNR value of 10 dB. In (a), all outcomes are included (Amp>0). Amp means the amplitude of the Gaussian peaks resulting from the**

275    **Gaussian-fitting process; the amplitude is normalized for respective range gates. In (b) and (c), the outcomes with Amp>0.5 and 0.8 are included in calculation, respectively.**

## 4 Tracing of Doppler velocities

With the Gaussian-fitted Doppler velocities obtained from the two approaches proposed in Sect. 2, in the following we describes the process to trace respective Doppler velocities of clear air and precipitation through the height.



### 4.1 Doppler profile of clear air

Starting from the lowest sampling gate, the two fitted Doppler velocities of greatest significance within ±2 m/s, that were determined respectively by the two proposed approaches, were averaged as an initial mean Doppler velocity ($v_1$) of clear-air motion. The respective amplitudes of the two fitted Doppler velocities were employed as the weights in the calculation of $v_1$. In case of no fitted Doppler velocities available in the first sampling gate, the second sampling gate was examined. This process could continue through several sampling gates until a value of $v_1$ was determined. In case no $v_1$ was obtained from the lowest 15 sampling gates (the number is user defined), the value of 0 m/s was assigned to $v_1$. Mostly, the first or several preceding sampling gates can provide the value of $v_1$.

Determination of mean Doppler velocity for the second sampling gate was based on $v_1$, i.e., by taking the fitted Doppler velocities within a velocity window of $-v_{dev}$ and $v_{dev}$ that centered on $v_1$, the weighted mean Doppler velocity was calculated as $v_2$. In the present calculation, $v_{dev}=3$ m/s. $v_2$ became the reference velocity of the next trace in the third sampling gate. In a similar fashion, a Doppler velocity profile for clear air can be obtained. However, it happened occasionally that there was no reference velocity for the present sampling gate when the calculation failed in the preceding sampling gate due to the diagnosis of Doppler velocity shear (described below). In case of this, the tracing process continued searching the existed mean Doppler velocity in the lower sampling gates. If no value was available in the 10 sampling gates lower than the present sampling gate, the searching process stopped.

In the tracing, the Doppler velocity shear is a crucial criterion. Assigning $v_n$, $v_{n-1}$, $v_{n-2}$, and $v_{n-3}$ to the mean Doppler velocities of the present and the lower three sampling gates, and $v_{shear}$ to the threshold of Doppler velocity shear, $v_n$ will be discarded if $|v_n-v_{n-1}|>v_{shear}$ or $|v_n-v_{n-2}|>1.5v_{shear}$ or $|v_n-v_{n-3}|>1.5v_{shear}$. For clear air, the value of 2 m/s was given for $v_{shear}$ in this study. Use of these Doppler shear criteria in tracing can increase the reliability of the traced Doppler profile although intermittent breaks in the profile may happen occasionally. It should be stated here that all values assigned for tracing are changeable, considering the atmospheric condition is variable. Once the tracing of clear-air motion finished, the fitted Doppler velocities within $-v_{dev/2}$ and $v_{dev/2}$, centering on the respective mean Doppler velocities of the sampling gates, were discarded in the tracing of precipitation velocity described below.

### 4.2 Doppler profile of precipitation

It is assumed that the Doppler speed of precipitation near the ground is apparently larger than that of clear-air motion in observation of using a vertical radar beam. Therefore, the initial mean Doppler velocity ($v_1$) of precipitation was calculated from the fitted Doppler velocities larger than 3 m/s. In case of no $v_1$ obtained in the lowest 15 sampling gates, 5 m/s was assigned to $v_1$. The process of tracing was the same as that for clear air. However, considering a wider range of Doppler velocities during the falling of precipitation, 3 m/s was given as the threshold of Doppler velocity shear ($v_{shear}$), which was larger than 2 m/s used for clear air. Notice that the fitted Doppler velocities used in tracing the Doppler profile of clear air had been discarded here.



Totally, 109 Doppler spectra were produced from the radar data shown in Fig. 3. Fig. 9 exhibits several typical tracing results, where the black and red curves are the traced Doppler profiles of clear air and precipitation, respectively. Original Doppler spectra are displayed in the background. Fig. 9a and b are two examples of pure air echoes, showing the tracing was

good, although the airplane echoes might interfere the tracing slightly, e.g., around the height of 7 km in Fig. 9b. In Fig. 9c, one more profile (red curve) was traced automatically for precipitation, showing the precipitation echoes occurred aloft and was much weaker than clear-air echoes. More successful tracings for both clear air and precipitation are exhibited in the panels (d)-(k) of Fig. 9, in which the strong interferences just above 2 km in the panel (j) have been treated properly in the tracing so that they were discarded in the profiling.

Two cases that were not traced completely are shown in the panels (l) and (m) of Fig. 9. The two Doppler spectra were similar to that in the previous time period (the panel (k)), but the Doppler profile of clear air broke around the height of 5 km and continued again since ~6 km. In the height interval of 5 and 6 km, the clear-air echoes were traced as precipitation echoes, and the weak precipitation echoes between 5 and 7 km, which were traced successfully in panel (k), were not identified. This was partly due to a large Doppler variation around 5 km altitude and partly owing to mixture of clear-air and precipitation

echoes, making the tracing difficult. The Doppler profile of clear air became continuous again later, as shown in the panels (n) and (o).

The two events in the panels (l) and (m) show that a large wind shear within a short range interval could make the tracing failed occasionally. In addition, it was sometimes difficult to distinguish clear-air and precipitation echoes when both types of signals had the Doppler velocities close to each other. For the present data which were collected in a severely convective

condition, only 2 events, i.e., the panels (l) and (m), out of 109 tracing results mistook the Doppler profiles of clear air and precipitation around the altitude of 5 km. In view of this, the success rate of tracing was really high. The 109 tracing results can be found via the supplement link. It is expected that the spectral parameters and tracing process validated in this study can be helpful for more scientific studies in the future. We provide a case study in the following section.

## 5 Observation campaign of convective precipitation aloft

As shown in Fig. 3, dynamic echoes were present in a height range between about 7 and 10 km in region 2. From the Doppler spectral tracing results shown in Fig. 9d-o, these echoes were characterized by very broad spectral width combined with intense upward air motion (sometimes larger than 5 m/s), and were identified as clear air echoes rather than precipitation echoes. Nevertheless, we observed that these echoes were followed by the presence of precipitation echoes from the supercooled droplets between about 5.5 and 7 km (the temperature of 0°C occurred at the height of about 5.7 km according to the radiosonde

data nearby; shown later). This feature seems to suggest that there existed hydrometeors in the height range of 7 and 10 km that were responsible for the formations of the supercooled droplets below 7 km. To verify this, the measurement of a collaborative dual-polarization microwave radiometer was examined to study the properties of the targets responsible for the echoes observed by the VHF radar.

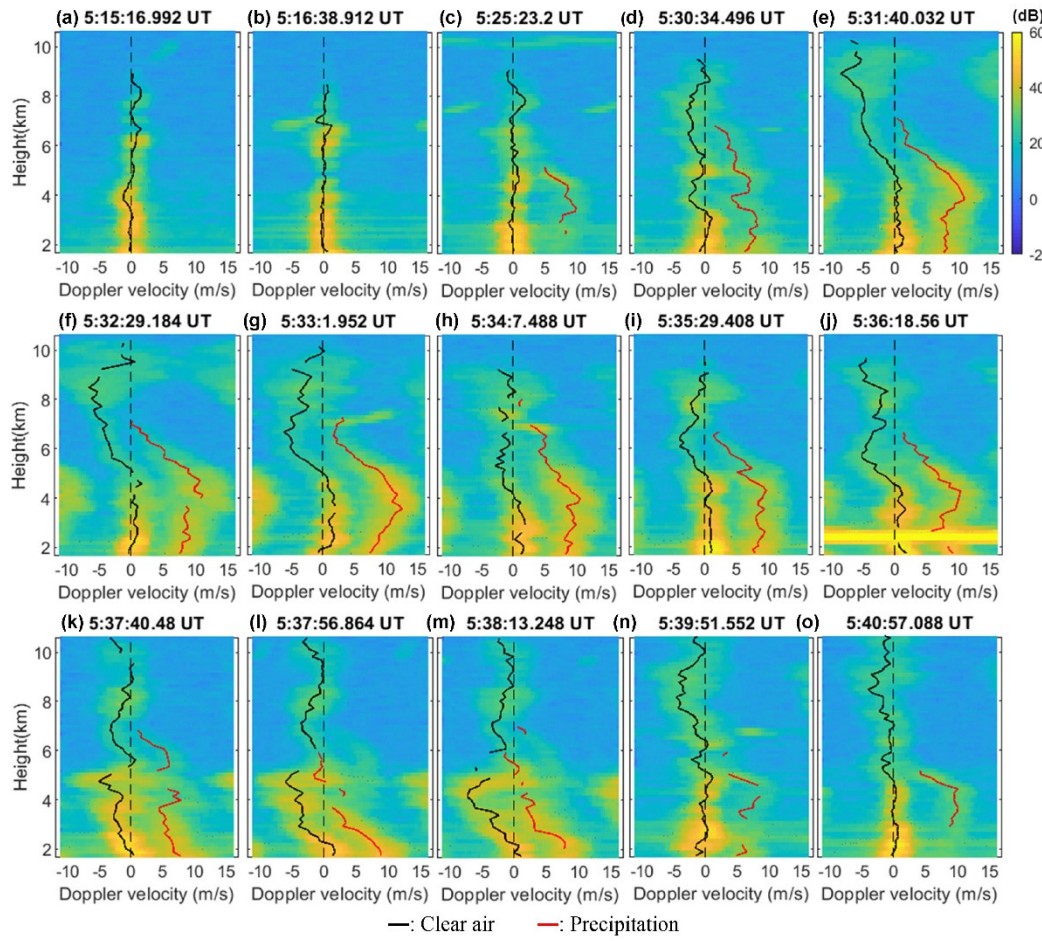

**Figure 9: Traced profiles of Doppler velocities for clear air (black curve) and precipitation (red curve), respectively. (a)(b) Clear-air echoes without precipitation. (c)-(o) Concurrence of clear-air and precipitation echoes.**

The dual-polarization microwave radiometer (model: RPG-4CH-DPR) was deployed on the campus of National Defence University (~200 m above sea level) that located at about 13.52 km southeast of the Chung-Li VHF radar site (~140 m above sea level). This ground-based radiometer can simultaneously detect the electromagnetic-wave radiations (i.e., thermal radiation) emitted from water vapour and liquid water particles (raindrop, cloud droplet, and supercooled water particle; exclusive of solid hydrometeors) with vertical and horizontal polarizations, thereby measure the so-called brightness temperatures at the two orthogonal polarizations. The difference between the brightness temperatures of vertical and horizontal polarizations, termed polarization difference (PD), can thus be estimated to determine the geometrical shape of liquid water particles. For a round liquid water particle, the PD will be very close to zero, and for a liquid water particle in a horizontally oblate shape, a negative PD is expected. In general, the bigger the liquid particle is, the flatter the liquid water particle will be, and a more negative value of PD is measured. Measurement from the frequency of 36.5 GHz has been examined for the purpose of this



study. It is noteworthy that according to Hou and Tsai (2019), the accuracy of measurement decreased gradually when the rain rate was higher than 20 mm/h, and became doubtful when the rain rate reached the value of about 60 mm/h. More descriptions

of the radiometer and measurement characteristics can be found in the study of Hou and Tsai (2019). Several key parameters of the radiometer are listed below. The receiving bandwidth is 400 MHz; range of the brightness temperature measurement is 0-350 K with a precision of 0.05 K; integration time (or data resolution) is 1 s; half power beam width (HPBW) of the antenna beam is 10.25° with a sidelobe level less than -30dBc; precision of PD measurement is ±0.25 K.

Fig. 10 depicts the configuration of experimental setup. The VHF radar beam was transmitted vertically, and the antenna

beam of the radiometer was steered in the northwest direction at an elevation and azimuth angles of 30° and 341.5°, respectively. Therefore, the observation zone of the radiometer at the distance of 13.52 km lay about 5 km to the east of the VHF radar site, and approximately in the height range between 6.33 and 9.57 km above the sea level of the VHF radar station. As a result, there was no overlapped field of view between the obliquely pointed radiometer beam and the vertically pointed VHF radar beam. Nevertheless, by taking account of finite beam widths, the separation of observational volumes between the radiometer

and the VHF radar was less than 5 km in the height range of 6.33 and 9.57 km. A separation of 5 km was not a considerable distance for the convective structure examined later that covered a horizontal region of over ten kilometers. Therefore, it was justified to compare the concurrent measurements of convective precipitation aloft made by the Chung-Li VHF radar and the radiometer.

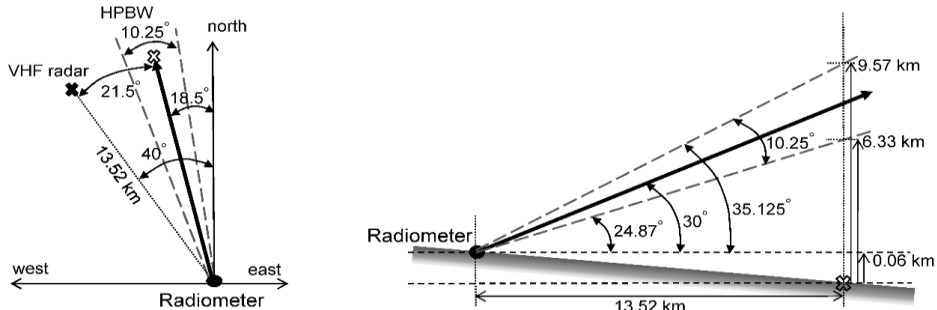


**Figure 10. Schematic plot of the radiometer and VHF radar locations (bullet and solid-cross symbols, respectively). (left) Antenna azimuth direction of the radiometer is 18.5º west of north, with a half-power beamwidth of 10.25º. (right) Antenna elevation angle of the radiometer is 30º, with a half-power beamwidth of 10.25º. The height interval of observation zone of the radiometer is between 6.33~9.57 km, above the sea level of the VHF radar.**


Fig. 11a provides five consecutive maps of meteorological radar reflectivity, which were trimmed from those released by the Central Meteorological Bureau of Taiwan. As shown, a strong front-like convective precipitation structure (or rainband), tilted in northwest-southeast direction, was moving from north to south at a phase speed of about 20 m/s. The rainband approached the VHF radar (black circle) and the radiometer (white circle) around 05:15 and 05:22 UTC, respectively, and left

the VHF radar after 05:45 UTC approximately. The rainband was a strong convective structure, according to the large and





prevailing upward Doppler velocities found in the VHF radar spectra shown in Fig. 9d-n. In the meantime, the PD shown in Fig. 11b increased from about -1.5 to around zero between 05:18 and 05:42 UTC and decreased quickly after 05:42 UTC, although the PD dropped intermittently around 05:31 UTC. Such variation in PD implies that the shape of liquid water particles varied mostly from horizontally oblate shape to almost round shape during the passage of the convective structure, which can

thus be speculated to be caused by an uplift of liquid water particles in the convective structure, i.e., the liquid water particles got more round during uplift.

In addition to PD, the dual-polarization microwave can also measure the total liquid water path (LWP), and distinguish liquid water of cloud (LWC) and liquid water of rain (LWR), as shown in Fig. 11c. LWP is a measure of the total mass of liquid water (i.e., liquid water content) in an atmospheric column above a unit surface area, and LWC and LWR denote

respective contributions from cloud droplets and liquid raindrops, in which solid water (snow, ice) is not included. It should be reminded that water vapour also contributes to the value of LWP. However, it is shown in Fig. 11c that LWP, LWC, and LWR varied in accordance with each other and LWP was approximately the sum of LWC and LWR.  In view of this, the contribution of water vapour to LWP was minor in the present event. Since LWP, LWC, and LWR decreased apparently between 5.4 and 5.6 h UTC, it implies a decrease of liquid water content in the observation volume of the radiometer in this

time period. The mechanism of causing a decrease of liquid water content is worthy of discussion on the basis of radar and radiometer observations during the passage of convective structure. We supposed that an interaction of ice crystals, water vapour, and supercooled water particles could be the candidate of the cause, as discussed below.

According to Pan-Chiao radiosonde measurement at 06:00 UTC  (Fig. 12), where is located at about 20 km northeast of VHF radar, the temperatures of 0°C, -10°C and -20°C occurred at heights of around 5.7 km, 7.6 km and 9.2 km, respectively.

Thus, it signifies that in the absence of the vertical air motion, the compositions of the hydrometeors in height ranges 5.7-7.6 km and 7.6-9.2 km were essentially dominated by supercooled water particles and the mixture of ice crystals and supercooled water particles in the cloud, respectively. However, in the presence of intense updraft, the updraft-carried water vapour and liquid water particles can significantly determine the growth of ice crystals and/or supercooled water particles above the 0°C isotherm level. The so-called Bergeron effect will proceed to transfer water vapour from supercooled water particles to ice

crystals due to the difference in relative humidity around supercooled water particles and ice crystals (Wang, 1997), leading to decrease of LWP. Because of the expense of supercooled water particles and/or water vapour during the Bergeron process, LWC and LWR decrease as well, as those seen around 05:30 UTC in Fig. 11c. As the ice crystals become larger, their weight increases and finally cause them to fall. During the fall of ice crystals, a second process, coalescence or accretion, continue increasing the weight of the falling ice crystals. Coalescence during the fall of ice crystals is a process that the supercooled

water particles collide and merge together, and then freeze on the surface of ice crystals (Khain et al., 2001; Korolev, 2007), resulting in the formation of graupel. The process of coalescence can also decrease LWP, LWC, or LWR.  We especially note that, in association with the updrafts, considerably intense turbulences that were characterized by the presence of very broad Doppler spectral widths (as large as 10 m/s or more) were observed in a height interval between about 7 and 10 km. Such intense turbulent condition can intensify the Bergeron effect and coalescence process.



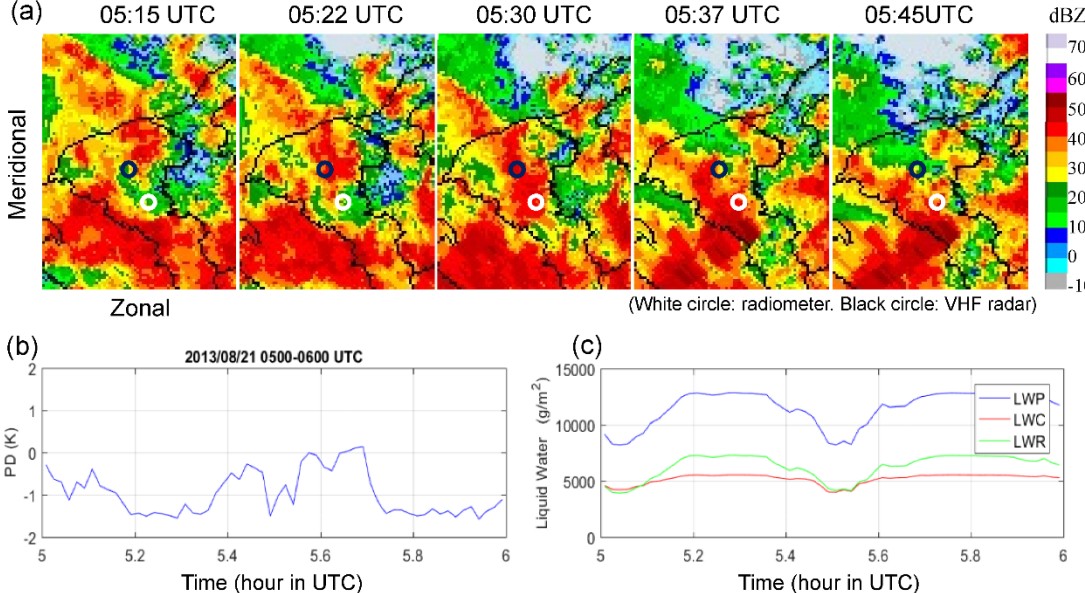


**Figure 11: Observations of weather radar and dual-polarization microwave radiometer. (a) Weather radar reflectivity maps (trimmed) between 05:15 UTC and 05:45 UTC, released from the Central Weather Bureau, Taiwan, on 21 August 2013. White circle: microwave radiometer. Black circle: VHF radar. (b) Temporal variation in polarization difference (PD) observed by 36.5 GHz microwave radiometer. (c) Temporal variations in total liquid water path (blue), liquid water of cloud (red), and liquid water of rain (green).**


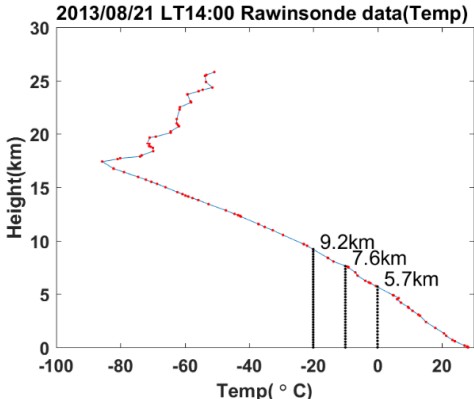

**Figure 12: Height-time temperatures (dots) observed by the Pan-Chiao Radiosonde Station. The temperatures of 0°C, -10°C and -20°C occurred at heights of around 5.7 km, 7.6 km and 9.2 km, respectively.**


Apparent falling of ice crystals and graupels started from the height of about 7 km, as shown in Fig. 9d-k. When the ice crystals and graupels felt below 0°C isotherm level (at around 5.7 km on the basis of the measurement of Pan-Chiao radiosonde at 06:00 UTC), they started to melt and became wet. It is expected that their backscatter will be about 5-15 dB larger than





those above the 0°C isotherm level (Liu, et al., 2000; Straka et al., 2000). As shown in Fig. 9e-j, a drastic increase in the radar backscatter of the falling ice crystals and graupels was about 5-10 dB in the height interval of 4-5 km, where was approximately below the 0°C isotherm level. This was coincident with the theoretical and previous experimental results (Liu, et al., 2000; Straka et al., 2000).

As ice crystals and graupels fall, their downward speeds are expected to increase. This feature was indeed observed clearly in Fig. 9e-j, where the downward speeds increased to 10 m/s at around the height of 4 km. It was also observed in Fig. 9 that

variations in the profiles of atmosphere and precipitation were coherent when the convective condition prevailed in the time interval of Fig. 9d-m. Therefore, it is evident that a convective atmosphere played a key role in the precipitation motion.

To verify further the depletion of liquid water content above the 0°C isotherm level, we employed the Weather Research and Forecasting model (WRF version 3) to simulate the distribution of liquid water for the precipitation environment investigated in this study, as shown in Fig. 13. Fig. 13a and b present, respectively, the simulated height-time densities of cloud water and

rain water at the grid point nearest to the VHF radar location, in which the time and vertical-range resolutions for the simulation were 0.5 h and 1 km, respectively. The initial field for the simulation was obtained from the National Center for Environmental Prediction (NCEP) reanalysis data at 00:00 UTC on 21 August 2013, and the horizontal resolutions were set as 6 and 2 km sequentially in the nested domain of calculation. As indicated by the red frames, both densities of cloud water (left panel) and rain water (right panel) drastically reduced above 6 km around 05:30 UTC, which was consistent with the decrease of LWC

and LWR shown in Fig.11c. This simulated result demonstrated further that the co-observational results of VHF radar and microwave radiometer were reasonable, that is, the composition of hydrometeor in the upper atmosphere (above the melting layer) changed due to strong updraft, Bergeron effect, and coalescence of supercooled water particles and ice crystals.

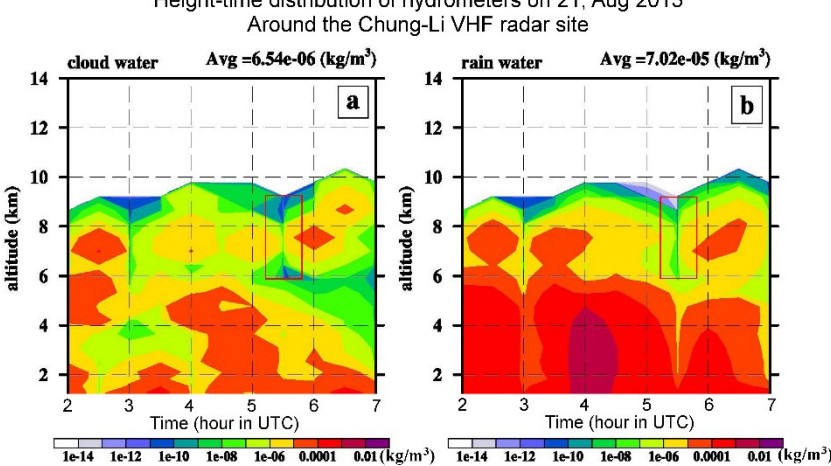

**Figure 13: Height-time distribution of liquid water simulated by the WRF model of Taiwan during the time interval of 02:00 and 07:00 UTC on 21 August 2013. Time and altitude resolutions are 0.5 h and 1 km, respectively. (a) Cloud water. (b) Rain water.**



## 6 Conclusions

One of the motivations of this study is to identify the respective Doppler velocities of clear air and precipitation automatically
when both types of signals coexist in the radar echoes. Two approaches have been developed for the purpose of the study, i.e.,
contour-based and peak-finding processes. To determine the major spectral centers or peaks for calculation of mean Doppler
velocity, several parameters such as SNR, contour level, peak amplitude, separation of peaks, and so on, have been considered.
The two approaches are complementary because not any of the two approaches can determine the major spectral peaks/centers
completely, at least for the present data set. Comparisons between the spectral parameters (mean Doppler velocity and spectral
width) of clear-air and precipitation echoes that appear simultaneously have been made. From the view of statistical distribution,
the results of the two approaches were in good agreement. Moreover, it showed that different numbers of data points in FFT
produced slightly different spectra widths, i.e., the smaller the FFT number was, the broader the spectral width could be. This
feature was more evident in rainy condition and could be attributed to the smoothing effect of using fewer FFT number and/or
more incoherent times of spectra.

With the reliable and complementary outcomes of the two approaches, tracing processes have been developed to determine
the respective Doppler profiles of clear-air and precipitation, in which the Doppler shear is one of the significant criteria in
tracing. In spite of occasional failure in tracing when the Doppler shear was too large, about 98% of the tracings for the present
radar data were acceptable. Because the parameters and criteria employed for determination of the spectral peaks/centers and
Doppler profiles are adjustable, it is possible to improve the tracings by adjusting these parameters and criteria properly for
various atmospheric conditions.

An application study using the traced Doppler profiles, radar echo powers, and spectral parameters above the melting layer
was also made for rainy and strongly uplifted condition during the period of a typhoon passage. This was achieved via the
collaborative observation using a dual-polarization microwave radiometer. Several information such as brightness
temperatures of vertical and horizontal polarizations of water, total liquid water path, liquid water of cloud, and liquid water
of rain yielded by the radiometer have signified the Bergeron effect, and coalescence/accretion process on formation of ice
crystal and graupel above the height of the melting layer under a strongly convective and turbulent atmosphere. The
circumstance of strongly convective and turbulent atmosphere was verified from the Doppler profile and spectral width of
VHF radar echoes. In addition, simulated calculations of range-time distributions of cloud water and rain water, obtained from
the Weather Research and Forecasting (WRF) model, also supported the scenario deduced from the radar and radiometer
observations. More efforts in the future are to test the Doppler profiling process for the radar data under different atmospheric
conditions and improve the criteria employed in the profiling. Moreover, collaborative observations of VHF atmospheric radar
and radiometer will be beneficial for further studies of dynamic atmosphere and precipitation.



Data availability. Data can be made available from authors upon request.

Competing interests. The authors declare that they have no conflict of interest.

Supplement link: https://drive.google.com/drive/folders/11vtU-Fh7s4YKcp2B-8d8XKF0hP4t8ICm?usp=sharing


Author contribution: Chen, J.-S. and Chu, Y.-H. designed the experiments and Tsai, S.-C. carried them out. Chen, J.-S. and Tsai, S.-C. developed the computing codes and performed the analysis. Tsai, S.-C. prepared the manuscript, and Chen, J.-S. and Chu, Y.-H. revised and finalized the manuscript.

*Acknowledgements.* The Chung-Li VHF radar is operated and maintained by the Department of Space Science and Engineering,
National Central University, Taiwan (ROC). The dual-polarized radiometer is owned by the National Defense University, Taiwan (ROC). We would like to thank Dr. C.-L. Su for helping on operation of the Chung-Li VHF radar. The radiometer and VHF radar data are obtainable by contacting the authors or the Chung_Li radar site (yhchu@jupiter.ss.ncu.deu.tw). This work was supported by the Ministry of Science and Technology, and China Medical University, ROC (Taiwan), under grants MOST109-2111-M-039-001 and MOST110-2111-M-039-001, and CMU109-MF-05, respectively.

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
