# Peer review of "Co-observation of strongly convective precipitation using VHF atmospheric radar and dual-polarized microwave radiometer during a typhoon passage"

_Atmospheric Measurement Techniques, 2021_

## Author Comment (AC1)

**AMT-2021-267**
**Responses to the reviewer's comments**

Reveiwer's comment:
This paper studies estimation methods of Doppler spectral parameters from clear-air and precipitation echoes by the Chung-Li VHF radar. Then radar observations of strong convective precipitation are compared with the data of dual-polarized microwave radiometer etc. when the typhoon Trami passed through Taiwan in August 2013. The proposed method shows a good performance, but this paper needs to be the revision.
Ans: We sincerely thank for the valuable comments from the reviewer. Point-by-point responses are given below. Paper title is revised, and moreover, the text and figures have been revised greatly according to the comments and suggestions from all reviewers. We add Figs. 3-5 to explain the processing steps in more detail, and remove the WRF results from the paper considering that the simulation results of WRF model are affected by many uncertain factors.
    Because the text is revised greatly, we highlight some parts of the text in blue color which response the major concerns of the reviewers.

Comments: L78 Contour-based approach
I cannot understand the benefits of this method because extending to the orthogonal spectral dimension with a Gaussian function does not increase any information. Please explain the theoretical background that shows the goodness of this method.
Ans: Thank for the question. Contour-based approach was initially applied to our 2D range imaging for identifying multiple targets' locations. We think it is also usable for identification of multiple spectral peaks after some modification of the process. After modeling a 2D spectrum from the original 1D spectrum, mean contour centers are estimated for respective spectral humps. The mean contour centers are still located at the original spectral axis, but may be not at the same spectral lines of the spectral peaks. This is different from the peak-finding approach that is also used in our study. The peak-find approach yields the spectral peak locations. In general, the results of the two approaches are consistent with each other, but discrepancy still exists sometimes in the spectral parameters (Doppler velocity and spectral width) retrieved from the two approaches, especially in multi-peaked spectra. The contour-based approach may provide the spectral parameters that are discarded by the peak-finding approach, and vice versa. Examples have given in Figs. 3-5 and 7 in the revised

manuscript.

With the outcomes of the two approaches, we develop a tracing process for multiple Doppler profiles. In the future, we can also fuse the outcomes retrieved from the already existed methods to raise the reliability of spectral parameters and Doppler profiling, but not in this study. We focus on showing the usability of the two approaches as well as the grouping, sifting, and Gaussian fitting processes introduced in this paper. These consequences can be references to the community.

Comment: L155 multiple carrier-frequency mode ...
Only the data from the carrier frequency of 52 MHz are analyzed. But the data from all the five frequencies can be used for incoherent integrations of Doppler spectra, because the frequency difference is only within 1%.
Ans: Thank you for the suggestion. It is true that the five frequencies can be used for incoherent integrations of Doppler spectra. However, considering that VHF radar observations use single frequency most of the time, the single-frequency data are tested in this study. We expect the methods and processes examined here can be applied to most of the data. However, this is a good suggestion in case multiple-frequency mode is operated in observation.

Comment: L170 Figure 3
The horizontal axes of Figures (a) and (b) should be aligned.
Ans: Revised. Fig. 3 is renumbered Fig. 6.

Comment: L245 Figure 6(a)
The altitude variation of the mean profile of the spectral width seems to be too small. I think that the spectral width is mostly determined by the beam broadening effect. Was the altitude change of the horizontal wind small at this time?
Ans: Thank you for the question. We add the horizontal wind observation of rawindsonde for discussion of this issue, as shown in Fig 15 in the revised manuscript. The rewindsonde is launched routinely at the location of ~25 km east-northeast of the VHF radar. A further study of beam broadening effect is beyond the object of this study. However, we have provided a discussion on this issue because the broadening of spectral width indeed exists due to either horizontal wind or some processes such as running mean, incoherent integration of spectra, and so on. Moreover, the discussion is also necessary for the application study given in Sect. 5. Please find the discussion given in Sect. 5.

In fact, considering that the broadening of spectral width exists commonly, we have performed the Gaussian fitting for the spectral peaks and contour centers with various fitting points, and the Gaussian curve with the smallest standard deviation is selected to estimate the spectral width. It is expected that such collected spectral widths can represent the characteristics of turbulence or precipitation in a greater degree.

Comment: L344 Figure 9
The time intervals are different, how did you choose? In Section 5, although Figure 9 is referred to as radar data, it is better to add the time-height plots for the entire period.
Ans: Thank you for the question. There are 109 maps of Doppler spectra produced from the radar data shown in Fig. 6 (in the revised manuscript). Due to limited space, however, only 15 plots are selected for discussion. They are typical spectra in the time periods when clear air, precipitation, and concurrent echoes dominate the spectra, respectively. It is difficult in a spaced-limited article to display the 109 plots of time-height Doppler spectra along with the Doppler profiles on the maps. We provide the whole maps of one day for readers, please find them via the hyperlink attached with this paper:
https://drive.google.com/drive/folders/11vtU-Fh7s4YKcp2B-8d8XKF0hP4t8ICm?usp=sharing

---

## Author Comment (AC2)

**AMT-2021-267**

**Responses to the reviewer's comments**

The manuscript "Co-observation of strongly convective precipitation using VHF atmospheric radar and dual-polarized microwave radiometer during a typhoon passage" by Tsai et al. presents roughly 30 minutes of data collected in 2013 by the VHF Chung-Li radar in Taiwan during the passage of the Typhoon Trami. An abundant part of the paper is spent on the description of the data processing while the case study is presented as an application of the measurement and processing technique. The observations are connected with the presence of an intense updraft which has brought substantial amounts of liquid water above the freezing level likely causing an intensification of the precipitation through various ice microphysical processes.

Ans: We sincerely thank for the valuable comments from the reviewer. Point-by-point responses are given below. Paper title is revised, and moreover, the text and figures have been revised greatly according to the comments and suggestions from all reviewers. We add Figs. 3-5 to explain the processing steps in more detail, and remove the WRF results from the paper considering that the simulation results of WRF model are affected by many uncertain factors.

Because the text is greatly revised, we highlight some parts of the text in blue color which response the major concerns of the reviewers.

GENERAL COMMENTS:

I had a hard time understanding what is the main message of the paper. The title of the paper suggests a measurement report where the main focus is the actual measurements and the combination of radar and radiometer observations. However, the abstract, the conclusions, and the vast majority of the paper focus on the techniques implemented for the processing of the VHF radar Doppler profiles. In particular, the introduction of the paper discusses only the state of the art in terms of Doppler peak finding algorithms without any mention of previous observations of strong, convective precipitation with VHF radars or the combination of radar and radiometer. In the introduction and conclusions, those measurements are discussed as an "Application example" of the profiling techniques. If it is really the author's intention to put the focus on the profiling technique I suggest modifying the title of the paper accordingly, alternatively, the manuscript should be significantly improved towards a more measurement-centric approach.

Ans: Thank you for the valuable comment and suggestion. Indeed, the initial objects of this study are to identify the spectral peaks of clear-air and precipitation echoes observed by VHF radar, estimate the spectral parameters effectively, and then develop a profiling process to trace the Doppler velocities of concurrent clear-air and precipitation echoes through the height. After that, we think we should present an application of the spectral parameters retrieved by the VHF radar to show the usability of the proposed approaches. We revise the title to match more the objects.

There have been a few collaborative studies with VHF/UHF radar and microwave radiometer in the literature, which aimed at atmospheric humidity or temperature profiling. We have cited these studies in the introduction.

Considering the importance that is given to the profiling technique in the introduction and conclusions of the paper and also the aim of AMT I will proceed in my review considering this as the main message of the paper.

The paper presents two distinct algorithms to identify the peaks in Doppler radar spectra. The introduction of the paper focuses on VHF and UHF radar techniques, for which, I must admit I am not an expert. However it seems to me that the introduction could benefit from additional considerations for similar techniques that have been developed for precipitation and cloud radars (1-100 GHz) which can be considered sensitive to hydrometeor particles only, but the mathematical foundations of peak finding in Doppler spectra are effectively the same. Also, it seems to me that the identification of air and hydrometeor peaks in VHF radars is a far easier task (with respect to peak identification in cloud radars). This is probably because of the selected application example. The respective mean peak location seems to be separated by roughly 10 m/s while their width is around 1 m/s avoiding the problem of the overlapping of the Doppler peaks. Honestly, I do not see why one cannot perform separation and tracing of the peaks using basic clustering algorithms in these cases.

Ans: Thank you for reminding us the achievements of precipitation and cloud radars. We have introduced these studies in the introduction.

For the VHF radars operated at around 50 MHz, the clear-air and precipitation echoes are comparable sometimes during heavy rain condition. Moreover, both echoes may not be separated distinctly through the height, especially near the bright-band altitude or light rain circumstance, where the precipitation velocity is small and can be close to that of clear-air echoes, leading to part overlap of multiple spectral humps. This fact causes the identification of both echoes difficult sometimes. Therefore, in the literature a few methods have been proposed for VHF radar to separate the Doppler profiles of concurrent clear-air and precipitation echoes. We believe that these existed methods, and that for cloud radars, have done good job and can work for most of circumstances. Tracing of respective Doppler profiles is another issue after proper separation of different types of echoes. It needs some processing and criteria to distinguish and trace the respective Doppler velocities through the height. As far as our knowledge, identification and tracing of concurrent clear-air and precipitation echoes are not always easy for VHF radar because of various conditions in the atmosphere, e.g., large wind shear, light or heavy rain, clutter and radio interferences, and so on. This is why there are many algorithms proposed for this task.

The motivation for the development of a new profiling technique is unclear to me. The introduction part stresses the unavailability of a unique algorithm that performs both the peak

identification and their tracing along the vertical, however, it is unclear to me how the algorithms that perform the tracing alone work without a peak identification. Also if established algorithms for peak separation and tracing exist what prevents one from combining them instead of developing a new one from scratch?

**Ans:** Sorry for unclear description of the motivation in the introduction. We have revised the text. To be honest, in our previous studies of concurrent clear-air and precipitation echoes with the Chung-Li VHF radar, it is highly relied on manual separation and adjustment of the two different echoes in the Doppler spectra, making the treatment of large data difficult and even impossible. In view of this, we expect to develop some kind of separation and profiling method/process for clear-air and precipitation echoes based on our own knowledge and experience, and then introduce the method and some ideas to the community; this fits the purpose of a research paper. There is something new in this article, for example, use of contour method, grouping and sifting of contour centers and spectral peaks, different FFT numbers, Gaussian fitting with various data lengths, proper selection of Gaussian fitting result, and so on. We believe that these processes and considerations are helpful to reduce manpower on identification of multiple spectral peaks, and improve the effectiveness of Doppler profiling.

Although there have been a few methods applicable to identification of multi-peaked radar spectra in the literature, we intend to find additional approach to identification of multiple peaks in the VHF radar spectra for our own radar and future study. However, it is not to negate the existed methods. It is surely that people can develop a profiling technique based on the existed separation methods. In the future studies, we can consider to fuse the outcomes of various separation methods to raise the reliability of Doppler spectral parameters and Doppler profiling.

The algorithms themselves are not presented very clearly. The core component of the algorithms relies on the implementation of a certain function in well-established commercial software (MATLAB, namely the contour and the peakfind functions) with not much addition from the authors. The original processing steps are presented in a way that is very difficult to interpret with little or no graphical examples. Moreover, the algorithm is implemented via several numerical or logical functions that rely on specific parameters which appear to be quite arbitrary. There is a consistent lack of explanation of the rationale behind the implementation choices which makes it difficult to properly judge the quality of the technique or to suggest improvements.

Ans: Thank you for pointing out the insufficiency of description of the algorithms. We have rewritten section 2 for the algorithms and explain the reasons for the specific parameters used in the process, such as different FFT numbers, Gaussian fitting numbers, velocity interval ($V_{interval}$), and so on. There are indeed considerations in use of these parameters. Moreover, we have added several graphics for interpretation of the processing steps (Figs. 3-5 in the revised manuscript).

Contour and peakfind functions in the commercial software just provide the initial locations of spectral peaks we need. Several steps (grouping, sifting, Gaussian fitting) are still required to identify and profile respective Doppler spectra. These treatments of the data can be regarded as

contribution from the authors. It is not as easy as we thought if one expects to profile the clear-air and precipitation echoes as accurate as possible for all circumstances.

Contour method has been applied to our previous study of 2D radar imaging for locations of multiple targets. We think it is also usable for identification of multiple spectral peaks after some post-processing of the contour centers in the modelled 2D spectra. Therefore, it is a new attempt in this field and can be introduce to the community. By giving proper values and criteria, qualified contour centers that represent the spectral humps can be obtained. The use of peakfind function is also parameter-dependent (peak numbers, prominence, and so on). According to user-defined parameters, it accepts or ignores the minor spectral peaks. Finally, grouping/clustering and sifting of these spectral peaks and contour centers are made to estimate and select the spectral parameters we need. These treatments may be not marvelous but work well, as demonstrated in the statistical comparison and Doppler profiling shown in Sect. 3 and 4, respectively.

The authors mention that the technique is automatic, but it seems also that several parameters have to be selected by the user and the application example proposed only provides a limited amount of data to prove the technique's performance. I have the feeling that the algorithm parameters have to be tuned according to the specific use case scenario in order to get the best performances. This makes the procedure of Doppler profiling a rather manual task and not an automatic one.

Ans: The "automatic" means without manual judgement of clear-air and precipitation spectral locations during computation. Once the initial values and criteria are given, the computation proceeds until the end of data set. However, this word has been revised in the text for avoiding mistook.

The atmosphere is highly changeable, and so it is difficult to fit all circumstances with a single set of parameters in the identification and profiling processes. However, we select the data set when a typhoon is passing the radar site, providing a wide range of weather conditions such as stable atmosphere, light and heavy rain, strong convective structure, and so on, although it is a limited amount of data. If the approach is workable for such varied circumstance, we can expect it also works mostly under moderate conditions. We have shown one-day profiling results, please find the hyperlink in the revised manuscript: https://drive.google.com/drive/folders/11vtU-Fh7s4YKcp2B-8d8XKF0hP4t8ICm?usp=sharing

Finally, the quality of the presentation and the graphics needs to be improved in order to facilitate the reading and understanding of paper content.

At the moment, it seems to me that the paper does not provide sufficiently innovative material to be published. Also, the quality of the presentation does not match the journal standards. I suggest reconsidering the paper for publication after a major revision, in particular, I would suggest the authors focus on the following aspects:

Ans: Thank you for the comment and suggestion. We will describe our contribution as clear as possible, and improve the quality of presentation and graphics in the revised manuscript.

1) Use a title that clearly conveys the paper main message

Ans: The title is revised.

2) Improve the quality of the graphics: Label subplots, avoid small fonts and plotlines (plot fewer lines instead), provide graphical (and theoretical) examples of the algorithm processing steps.

Ans: We have added some graphical examples of the processing steps, as shown in Figs. 3-5. In addition, some plots can be
 enlarged to avoid small fonts and plotlines. Labels are also provided for all figures.

For some plots, however, several curves are shown in the same panel for comparison. We have tried to describe the meaning of these curves or lines in the Figure Caption and in the text as clearly as possible.

3) Explain the algorithm design choices with their scientific background. Why is a certain step performed? Why does a certain parameter have that precise numerical value? Why not use already developed processing methods?

Ans: We have rewritten Sect. 2 to explain the processing steps as logistically as possible, including the reasons of why the numerical values are used, for example, FFT number, Gaussian fitting number, velocity interval ($V_{interval}$), and so on.

We develop the processes of multipeak identification and Doppler profiling based on our own knowledge. As a goal of the research work, we expect to introduce new methods, processes, and ideas to the community. This is not to negate the already developed processing methods in the literature. It is surely that people can develop a profiling technique based on the existed separation methods. It is possible to fuse the outcomes of the existed methods to improve the estimate of spectral parameters and Doppler profiling in the coming study. If we decide to use just the already developed processing methods, the research should be oriented to be more scientific. However, this is not our initial goal of this study.

4) If possible, provide validation of the results against independent measurements.

Ans: Thank you for the suggestion. The Chung-Li VHF is a unique radar which can observe the clear-air turbulence and precipitation simultaneously around the observational area. It is difficult to compare all spectral parameters with other instruments at different frequency bands. We have distrometers nearby and also have a radiometer (~13.5 km distance) that are operated jointly in the observation. Parts of these independent measurements have been employed in the study, as present in Sect. 5.

We used to compare the wind measurement with that detected by the rawinsonde located at the distance of ~25 km from the VHF radar site. The rawinsonde data have also been discussed in this study, as given in Sect. 5.

5) Explain better the scientific context of the provided application example. What is the scientific value of the measurements?

Ans: Thank you for the comment. Section 5 shows an application example with the Doppler spectral parameters obtained from our VHF radar. The measurement of a microwave radiometer is jointly used to discuss on the Bergeron effect and coalescence process above the melting layer. A few collaborative studies with VHF/UHF radar and microwave radiometer existed in the literature, however, they aimed at atmospheric humidity or temperature profiling. In our study, we discuss the dynamic variation of hydrometeors above the melting layer, which has not been discussed before, as far as we know. We believe that this application study contributes some new ideas to the community when people use VHF/UHF and microwave radiometer jointly. The text in Sect. 5 is revised.

DETAILED REVIEW:

L13 - What does it mean that the peak locations are redundant?

Ans: The peak locations determined from the contour-based and peak-find approaches may not be at the same spectral position, but indicate the same type of echoes or spectral hump. This sentence has been revised in view of confusing the reader.

L52 (and many other places) - The authors use the term "Doppler parameters" many times without defining it. What are those parameters?

Ans: We use Doppler parameters two times, and spectral parameters many times. We unify it as spectral parameters, which means "Doppler velocity and spectral width" in this paper. This will be stated at the first place it appears and when it is necessary to clarify again.

L71 - Why was it necessary to derive 3 Doppler spectra per sample using 3 different FFTs?

Ans: Thank you for the question. We have added the reason for using different FFT numbers in Sect. 2.1, the first paragraph.

L80 - The only motivation I can imagine for constructing such a 2D field is that the contour function works with 2D fields only. Is this the case?

Ans: Yes, the contour function works with 2D field only. It works for the modelled 2D spectra. We have tried to show how to use the contour method in identification of multiple peaks in the 1D radar spectra.

The 2D functions are symmetrical around the "true" frequency axis?

Ans: Yes. A graphical example is shown in Fig. 2.

Why standard deviation is specifically Nfft/8? Standard deviation should be in the same measuring unit as the axis, which means that 8 has units of seconds, right?

Ans: Sorry for the confusion. It is the number of spectral lines, not time unit. As shown in Fig. 2, the extended dimension is denoted Doppler velocity, not second. The number of spectral lines assigned to the standard deviation is changeable, it does not alter the contour center except a tiny error from computation. We have tried to describe this point more clearly in the first step in Sect. 2.1.1.

L90 - n is renamed Nlevel. Should use only one variable name
Ans: Revised

Fig 2 - the subplots should have labels. The second subplot must have a much larger axis, it is not possible, at the moment to appreciate the various contour lines. Also, other figures display spectra as a function of velocity and not frequency, it is better to be consistent and velocity holds a more geophysical significance.
Ans: Revised.

L103:105 - I still think that the gaussian functions are symmetrical, so the centers should not be shifted at all.
Ans: Yes. The contour centers are not shifted except a tiny error from computation. We have rewrite the text to confirm it.

L110:113 - I did not understand this phrase. I think this is a very central point since it should explain how to go from a contour map to some peak locations. I think that the explanation could benefit from some visualization of either real data or example data that could show how this process actually works. Also, it would allow discussing some deficiencies of the method. As an example right peak of Fig 2 (first panel) will likely be highlighted by two separate contours but in reality, it seems to me like only one peak with a lot of noise; thus the contour method would identify only one of the two subpeaks shifting its estimated mean Doppler velocity.
Ans: We revise Fig. 2 and add Figs. 3-5 to explain the steps in the contour method. Multiple peaks ride on a spectral hump sometimes, like the example shown in Fig. 2. Some of these peaks may be grouped into one mean center or discarded after sifting and Gaussian fitting.

L119 - Are the 5,7,9,13 lines separated by the resolution of the Doppler spectrum? Isn't this limiting the estimation of the spectral width? Is the range of spectral widths that one can estimate dependant on the spectral resolution (number of FFT points) then? I think that this is rather arbitrary and in nature, the width of a spectral peak depends on a variety of factors, especially for precipitation peaks the algorithm choices made here seem very restrictive. Also, what is the reason behind the choice of selecting the Gaussian fit with the minimum standard deviation?
Ans: In fact, we use five numbers of spectral lines: 5, 7, 9, 11, 13, in the Gaussian fitting process. This written mistake is corrected. The reasons for use of different numbers in Gaussian fitting are: (1) The fitting with a specific number may not success always,

depending on the distribution of spectra lines around the spectral peak/mean center. We expect that at least one of the fittings is available. (2) The spectral width varies with some factors including the fitting number and spectral resolution.

Among the multiple fitting results, we select the one with minimum standard deviation as outcome. The reasons for such selection are: (1) finite data length, which results in a Sinc function in frequency domain, and incoherent integration and running mean of spectrum tend to broad the spectral width; (2) beam broadening effect associated with horizontal wind on the spectral width exist commonly. It is expected that such collected spectral widths can represent the characteristics of turbulence or precipitation in a greater degree. We mention these reasons in Sect. 2.3 in the revised manuscript.

L137 - Why is this passage necessary? Why the findpeaks function does not find representative mean locations?
Ans: Thank you for the question. It forces us to check our codes in peak-finding approach one more time. We found the process of mean estimate of each spectral peak is in fact not performed although it was included in the codes. We ignored this process in the final version. Examples of peak-finding results are shown in Fig. 3 in the revised manuscript, and the description of this approach is given in Sect. 2.2.2.

L130:140 Again, this explanation could be improved with some graphical example on a real or fictional spectrum.
Ans: Thank you for the suggestion. Graphical examples are provided in Figs. 3-5.

Paragraphs 2.1 4) and 2.2 2) are not part of the respective peak finding algorithms and are redundant. They can be joined in one subsection. Logically, section 2 should be structured as follows:
2 Doppler spectra analysis
2.1 Preprocessing
2.2 Peak identification
2.2.1 Contour method
2.2.2 Findpeaks method
2.3 Estimation of mean peak velocity and width
Ans: Thank you for the suggestion. Sect. 2 has been re-structured, and graphical examples are provided to describe the processing as clearly as possible.

Fig 3 x-axes should match
Ans: Revised. The revised figure number is 6.

Fig 4 a) and b) It is very hard to distinguish between the black/red/open circles. DIstinguishing lines is even harder. Maybe this figure could be reduced to a couple of specific portions as it is done in panels c) and d)

Ans: Thank you for the suggestion. However, the purpose of Fig. 4a and b is to show a whole picture of the three outcomes, demonstrating the outcomes are consistent with each other in general. A detailed comparison of examples is given in Fig. 4c, which can distinguish the difference between the outcomes. We can enlarge the plots for an easier inspection. The figure is renumbered Fig. 7.

Fig 5 Again, not really readable. Focus on height regions or reduce the vertical resolution

Ans: We can enlarge the plots for an easier inspection. The figure is renumbered Fig. 9. Again, we demonstrate here the whole picture of the outcomes, and the statistical comparisons of the results are provided in Fig. 10. We can enlarge the map in the paper, although it occupies more article space.

Section 3 - Rather than a validation of the method this section shows an application example. There is no independent verification of the obtained results performed. The analysis shows a rather consistent output between the two methodologies. I am not sure if one should take this as a confirmation of the validity of the methodologies applied. Again, the used example case seems easy enough to not pose too many challenges, however, I do not see a clear validation of the methodologies, a comparison of the respective benefits and flaws, a discussion of the advantages of these methods with respect to already established ones.

Ans: The section title is revised "3. Statistical characteristics of spectral parameters" because in this paper we do not intend to compare respective benefits and flaws of the already existed methods with our approaches. We believe that the already established methods in the literature are workable and may have different considerations. The existed methods have been cited as far as possible in the introduction.

We propose our own approaches for readers' reference. Several figures have shown the usability of the approaches (e.g., Figs. 2-5, 7). In a practical operation, it is surely encouraged to include diverse outcomes of different methods in the Doppler profiling as well as deepgoing researches, but not in this study. One of the reasons is that we have not yet established the processes of the already existed methods, and the other reason is that we expect to pay more effort on the present methods and discuss some issues in a limited article space that have not been or rarely discussed before, for example, different FFT and Gaussian-fitting numbers, collaborative studies of hydrometeors with VHF radar and radiometer for the height range above the melting layer, and so on.

L298 - Why 1, 1.5, and again 1.5 times Vshear as thresholds?

Ans: These are experience values which yield good profiling results for the strongly convective atmosphere with heavy rain. As mentioned in the article, the strongly convective

atmosphere with heavy rain provides an excellent test-bed of identification approaches and profiling process. In case the parameters assigned to the process are usable for such convective atmosphere during a typhoon passage, probably they could be suitable for other moderate situations. Certainly, we cannot expect the selected parameters validate for all extremely varied weather, for example, the wind velocity may not change monotonically along the height. Moreover, considering that interference or other targets such as airplanes could also vary the Doppler velocity abruptly, which makes us to use a finite value of $V_{shear}$ for avoiding mistook of non-atmospheric targets in profiling.

In view of diverse conditions in the atmosphere, we prefer some values assigned for tracing and identification changeable. Flexibility of these values may provide us a better result, and can also reduce a lot of manpower on judgement of clear-air and precipitation once the computing process starts.

L300 - The fact that all parameters are changeable to adapt to atmospheric conditions makes me think that the method is not really automatic, which is in contrast with what is stated in the conclusions.

Ans: The "automatic" means without manual judgement of clear-air and precipitation spectral locations during computation. Once the initial values and criteria are given, the computation proceeds until the end of data set. However, this word has been revised in the text for avoiding mistook. More reasons for use of changeable parameters in the process can refer to the response of preceding question.

L350 - I do not agree with this statement. Any object emits radiation at every wavelength. The radiometer will detect the radiation coming from any emitting substance and not absorbed along the path. It is true that ice emissivity is lower than the one of water vapor or liquid water and might be negligible, however, it is not null.

Ans: It is true that the radiometer detects the radiations coming from any emitting substance and not absorbed along the path. Therefore, the radiation frequency around the operation frequency (bandwidth 400 MHz) will be collected. However, the contribution from atmospheric molecules and solid hydrometeors is tiny and so can be ignored, as compared with the emissivity of water vapour and liquid water. Sorry for incomplete description here. The description is revised.

L390:402 - The measuring principle of the radiometer is not discussed enough. In my opinion, it is not sufficient at this stage to only cite Hou et al work to describe the radiometer. The authors should clearly state what are the measuring frequencies of the DPR (I guess 4CH means 4 channels and thus all 4 are used for the LWP retrieval?). The quantities LWP, LWC, and LWR are taken as basically observational truth without any discussion about the measurement uncertainties. In particular, it would be interesting to know how the radiometer estimates the content of precipitating raindrops and cloud water. Also, as far as I remember, radiometers do not work really well when they are covered by water which I would think it is the case considering the

intense precipitation event. How does that impact the measurements?

Ans: Yes, the dual-polarization microwave radiometer (model: RPG-4CH-DPR) has four channels working in horizontal and vertical polarization at 18.7 GHz and 36.5 GHz, and LWP is retrieved from the 4 channel measurements. We revise the text in Sect. 5 and provide more information of the radiometer. Since microwave radiometers have been employed for decades and plenty of studies have been made to demonstrate exclusively the information and physical background of measurements with microwave radiometers, we do not give again too detailed information of the radiometer but focus on the joint discussion on the measurements of radiometer and VHF radar. We have found many exclusive studies of microwave radiometers and cite some of them in the paper, in addition to Hou and Tsai.

To prevent the radiometer antenna from being covered by water, a hydrophobic coating was established around the receiving antenna. It is useful for preventing the water from covering the antenna.

L359 - According to Fig 3 the rain rate Around 05:30 is close to 50 mm/h which makes it very likely that the signal observed in the PD and LW time series is just a result of the decreased accuracy of the radiometer stated in this line

Ans: Thank you for the question. The rain rate shown in Fig. 6 (original Fig. 3) was detected by a disdrometer located at the VHF radar site. We have checked the observations of two more rain gauges located at Dasi and Bade, where are closer to the radiometer and also closer to the observational path of the radiometer, as indicated in Fig. 14a (x shapes). The two rain gauges provide the rain rates hourly, and show the values of 3.5 and 19.0 mm/h, and 6.5 and 17.0 mm/h, respectively, at 05:00 and 06:00 UTC. The two sets of rain rates are much smaller than the rain rate measured by the disdrometer at the VHF radar site. Moreover, although the study of Hou and Tsai (2019) shows the positively linear relationship between brightness temperature and rain rate decrease gradually when the rain rate is higher than 20 mm/h, and become doubtful when the rain rate reaches the value of about 60 mm/h, the brightness temperatures still increase gradually between the rain rates of 20 and 60 mm/h. In view of these observations, we suppose that the measurements of the radiometer are still reliable.

L392 - 402 I am not sure I can understand the author's interpretation of the measurement. According to the observation geometry the radiometer observers only the atmosphere above the melting layer at the radar distance. This means that any drop of LWP is more likely to be connected to decreases of liquid content below (and hence closer to the radiometer). Moreover, in presence of a strong updraft, I expect the LWP to increase above the melting layer, or? I think the authors should explain better their interpretation.

Ans: According to the weather radar reflectivity maps shown in Fig. 14a, the rain band was approaching and then covered the area between VHF radar and radiometer during the time interval of observation. Therefore, the liquid water content in the observation volume of the radiometer is expected to increase gradually until ~05:37 UTC. However, we observe that LWP,

LWC, and LWR decreased gradually and then recovered from the valley during the time period of 5:24 and 5:36 UTC, it implies a general decrease of liquid water content in this time period. In view of this, the mechanism responsible for decrease of liquid water content are discussed. We supposed that an interaction of ice crystals, water vapour, and supercooled water particles above the melting layer could be the candidate of the cause.

In the presence of intense updraft, the updraft-carried water vapour and liquid water particles can significantly determine the growth of ice crystals and/or supercooled water particles above the $0°C$ isotherm level. The so-called Bergeron effect will proceed to transfer water vapour from supercooled water particles to ice crystals due to the difference in relative humidity around supercooled water particles and ice crystals, leading to decrease of LWP. As the ice crystals become larger, their weights increase and finally cause them to fall. During the fall of ice crystals, a second process, coalescence or accretion, continue increasing the weights of the falling ice crystals. Coalescence during the fall of ice crystals is a process that the supercooled water particles collide and merge together, and then freeze on the surface of ice crystals, resulting in the formation of graupel particles. The process of coalescence can decrease LWP, LWC, or LWR, too. These discussions have been given in Sect. 5.

Fig13 (and discussion around the WRF model) - The authors say at L446 that the time resolution of the simulations is 30 minutes however in the figure the drop in liquid water content seems to be connected to a much higher time resolution; the colors vary much faster, what is the reason for that? Also, do the WRF cloud-microphysics implement the processes (like secondary ice generation) that are speculated to be responsible for the observed features? Finally, is the time of the forecast (5.5 hours) enough to be out of the expected spin-up time of models?
Ans: Consider that the simulation results of WRF model are affected by many uncertain factors (e.g., intense updraft, topography, time of the forecast, and those raised by the reviewer), we decide to remove the WRF results from the paper.

L400 - I LWC decreases shouldn't also the radar echo intensity decrease? It is difficult to judge from the plots of Fig 9 what is the average intensity of the signals. I think one can add to the plots of fig 9 a secondary x-axis that shows the profiles of radar reflectivity.
Ans: Thank you for the suggestion. However, the VHF radar echoes come from clear-air turbulence and precipitation (if exist). Decrease of LWC does not indicate the VHF radar echo intensity will decrease as well. This is different from that of cloud radar.

L462 - I do not understand why the authors say that none of the two approaches can identify the peaks completely. What is the meaning of the word "completely" here? Also in the discussion of the methods, it is stated that the two approaches give the same results, this is in contradiction with what is stated here where the approaches are described as complementary
Ans: Sorry for the confusion due to unclear description. The text is revised. The outcomes of the two approaches are consistent with each other in general. However, discrepancy in the spectral

parameters (Doppler velocity and spectral width) still exists sometimes, especially in multi-peaked spectra. The contour-based approach may provide the information of spectral humps that are discarded by the peak-finding approach, and vice versa.

MINOR POINTS:

L17 - "in the tracing process" is redundant
Ans: The abstract is revised.

L19 - even WHEN the atmosphere was disturbed severely
Ans: The abstract is revised.

L21 - The verb "to signify" does not really make too much sense to me in this phrase
Ans: The abstract is revised.

L33 - preposition "in" is incorrectly used
Ans: The sentence is revised. Use "from" instead of "in".

L43 - "more accurately" or "in a more accurate way"
Ans: Revised.

L50 - What is the meaning of "exclusive" here? What has to be excluded?
Ans: The text is revised.

L71 – points
Ans: Revised

L72 - The word "respectively" should not be here. Respective to what?
Ans: Revised. The structure of Sect. 2 is modified.

L73 - 64 points?
Ans: Revised

Fig 1 - This figure is not really necessary to me. Both algorithms are very linear with no forks or loops. It does not add anything to what is already in the text.
Ans: Some readers may prefer a brief flowchart to text only, according to some comments on our previous studies. Let us remain it here for diverse readers.

L108 - the term "velocity interval" is introduced without explanation

Ans: Thank you for reminding this point. The reason for using the value of 1.4 m/s for velocity interval has been provided in Sect. 2.2.1, 3), embedded in the third paragraph.

L111 - the separation ... IS larger
Ans: The text is revised.

L223:224 A further comparison ... is shown
Ans: Revised. The comparison shown in the original Fig. 7 is moved to Sect. 3.1, and Fig. 7 is renumbered Fig. 8.

L226 - the word "compared" should not be here
Ans: Deleted.

L229 - "no matter what the methods were used, the estimated spectral parameters were in good agreement,"... in good agreement with what?
Ans: The text is revised, as given in the final paragraph of Sect. 3.1.

L305 - What is it meant by the term "apparently"?
Ans: It is deleted.

Fig.13 - colorbar has no label for the shown quantity
Ans: This figure has been removed.

Fig 11 (but also Fig.3) - the time axis is given in floating-point hours. It would be much better to have consistently hours and minutes as the format of times throughout the paper. Here is critical for the examination of the simultaneous measurements
Ans: Thank you for reminding us this point. We have revised the format of time axis as hours and minutes.

A FEW REFERENCES FOR PEAK FINDING ALGORITHMS IN CLOUD RADARS
Radenz, M., Bühl, J., Seifert, P., Griesche, H., and Engelmann, R.: peakTree: a framework for structure-preserving radar Doppler spectra analysis, Atmos. Meas. Tech., 12, 4813–4828, https://doi.org/10.5194/amt-12-4813-2019, 2019.
Kalesse, H., Vogl, T., Paduraru, C., and Luke, E.: Development and validation of a supervised machine learning radar Doppler spectra peak-finding algorithm, Atmos. Meas. Tech., 12, 4591–4617, https://doi.org/10.5194/amt-12-4591-2019, 2019.
Ans: Thank you for the suggestion. The two papers are cited in the paper.